# Peer review of "Artificial Intelligence on Food Vulnerability: Future Implications within a Framework of Opportunities and Challenges"

_societies, doi:10.3390/soc14070106_

Round 1
Reviewer 1 Report
Comments and Suggestions for Authors
The paper discusses the role of artificial intelligence (AI) in addressing global food vulnerability, emphasising both the opportunities and challenges it presents. It reflects on Stephen Hawking’s warnings about the potential dangers of AI, particularly in relation to food systems. The paper emphasises the crucial role of ethical oversight in the development and application of AI technologies, particularly in the context of food production and distribution. The main thesis of the paper is that artificial intelligence (AI) has transformative potential to address global food challenges, but it also carries significant risks that require careful oversight and ethical consideration. The paper explores how AI can optimise agricultural practices, enhance food security, and personalise nutrition, while emphasising the need to address ethical concerns and potential dangers of superintelligent AI. The author of the article employs a qualitative approach to explore the implications of artificial intelligence (AI) on food security. He or she analyses the challenges and opportunities presented by AI in various fields, emphasising the global food reality. The study also considers ethical oversight and the balance between technological progress, the preservation of local wisdom, and environmental sustainability. The author of the article employs a qualitative approach to explore the implications of artificial intelligence (AI) on food security. They analyse the challenges and opportunities presented by AI in various fields, emphasising the global food reality. The study also considers ethical oversight and the balance between technological progress, the preservation of local wisdom, and environmental sustainability. Key methods include examining AI's impact on food security through a contemporary lens, reviewing existing research on AI, food vulnerability, and related ethical considerations; discussing specific scenarios in which AI can optimise food production and distribution and evaluating the potential risks and ethical implications of AI in the context of food security. The paper appears to present a unique analysis, combining Stephen Hawking's warnings about AI with the specific issue of food vulnerability. It provides a modern viewpoint on AI's potential to address global food shortages and sustainable food production. The paper highlights the need for ethical oversight in AI's application to food systems. It includes a qualitative approach to assess AI's role in improving food security and managing food crises. The goals, methods, and conclusions are woven together to present a narrative that AI can be a powerful tool for addressing food vulnerability if managed with careful consideration of its potential risks and ethical implications.
Author Response
Comments:
The paper discusses the role of artificial intelligence (AI) in addressing global food vulnerability, emphasising both the opportunities and challenges it presents. It reflects on Stephen Hawking’s warnings about the potential dangers of AI, particularly in relation to food systems. The paper emphasises the crucial role of ethical oversight in the development and application of AI technologies, particularly in the context of food production and distribution. The main thesis of the paper is that artificial intelligence (AI) has transformative potential to address global food challenges, but it also carries significant risks that require careful oversight and ethical consideration. The paper explores how AI can optimise agricultural practices, enhance food security, and personalise nutrition, while emphasising the need to address ethical concerns and potential dangers of superintelligent AI. The author of the article employs a qualitative approach to explore the implications of artificial intelligence (AI) on food security. He or she analyses the challenges and opportunities presented by AI in various fields, emphasising the global food reality. The study also considers ethical oversight and the balance between technological progress, the preservation of local wisdom, and environmental sustainability. The author of the article employs a qualitative approach to explore the implications of artificial intelligence (AI) on food security. They analyse the challenges and opportunities presented by AI in various fields, emphasising the global food reality. The study also considers ethical oversight and the balance between technological progress, the preservation of local wisdom, and environmental sustainability. Key methods include examining AI's impact on food security through a contemporary lens, reviewing existing research on AI, food vulnerability, and related ethical considerations; discussing specific scenarios in which AI can optimise food production and distribution and evaluating the potential risks and ethical implications of AI in the context of food security. The paper appears to present a unique analysis, combining Stephen Hawking's warnings about AI with the specific issue of food vulnerability. It provides a modern viewpoint on AI's potential to address global food shortages and sustainable food production. The paper highlights the need for ethical oversight in AI's application to food systems. It includes a qualitative approach to assess AI's role in improving food security and managing food crises. The goals, methods, and conclusions are woven together to present a narrative that AI can be a powerful tool for addressing food vulnerability if managed with careful consideration of its potential risks and ethical implications.
Response:
Dear reviewer, thank you very much for your comments.
Reviewer 2 Report
Comments and Suggestions for Authors
Thank the authors for submitting the manuscript titled “Artificial intelligence on food vulnerability: future implications within a framework of opportunities and challenges”. This paper aims to understand the implications of AI in food security and explore the challenges and opportunities of AI applications with an emphasis on global food security. Although there are some novelties, I still have some comments for the authors to improve this manuscript.
1. Line 39, Page 1: “[3] report 27”, to start a new sentence, please use the authors’ names to represent a study. The same issue on Line 95, Page 3 “[25] warned that”.
2. Section 5.2: I suggest that the sub-title of each paragraph can be removed.
Comments on the Quality of English LanguageThe quality of English Language is good.
Author Response
Comments:
Thank the authors for submitting the manuscript titled “Artificial intelligence on food vulnerability: future implications within a framework of opportunities and challenges”. This paper aims to understand the implications of AI in food security and explore the challenges and opportunities of AI applications with an emphasis on global food security. Although there are some novelties, I still have some comments for the authors to improve this manuscript.
- Line 39, Page 1: “[3] report 27”, to start a new sentence, please use the authors’ names to represent a study. The same issue on Line 95, Page 3 “[25] warned that”.
- Section 5.2: I suggest that the sub-title of each paragraph can be removed.
Response:
Dear reviewer, thank you very much for your comments, your remarks have been addressed, they can be seen in the manuscript highlighted in red. In addition, the subtitles have been removed from the manuscript.
Reviewer 3 Report
Comments and Suggestions for Authors
After reading the manuscript "Artificial intelligence on food vulnerability: future implications within a framework of opportunities and challenges", I have a few comments.
Line 5: It would be helpful to briefly explain "the lens of Stephen Hawking" in the introduction, so the reader knows what it refers to.
Line 19: Please remove Fanzo.
Line 178: Please remove Haelein and Kaplan.
Line 203: Also, delete Soori et al.
Lines 205-208: Could you review the writing style in this section?
Line 451: Please specify the source of table 1.
The study explores the challenges and opportunities of AI in different fields, with a focus on global food reality (Lines 8-9). However, the content remains quite general with little emphasis on food issues. The conclusion, while not mentioning food issues, fits well with the rest of the manuscript. In the conclusion (lines 526-528), the authors discuss "the multifaceted nature of AI, its impact on various sectors," which aligns with the content of the manuscript. Therefore, I suggest that the authors remove "Food" from the title and come up with a title that reflects AI's impact on society in general.
Comments on the Quality of English LanguageComments
After reading the manuscript "Artificial intelligence on food vulnerability: future implications within a framework of opportunities and challenges", I have a few comments.
Line 5: It would be helpful to briefly explain "the lens of Stephen Hawking" in the introduction, so the reader knows what it refers to.
Line 19: Please remove Fanzo.
Line 178: Please remove Haelein and Kaplan.
Line 203: Also, delete Soori et al.
Lines 205-208: Could you review the writing style in this section?
Line 451: Please specify the source of table 1.
The study explores the challenges and opportunities of AI in different fields, with a focus on global food reality (Lines 8-9). However, the content remains quite general with little emphasis on food issues. The conclusion, while not mentioning food issues, fits well with the rest of the manuscript. In the conclusion (lines 526-528), the authors discuss "the multifaceted nature of AI, its impact on various sectors," which aligns with the content of the manuscript. Therefore, I suggest that the authors remove "Food" from the title and come up with a title that reflects AI's impact on society in general.
Author Response
Comments:
After reading the manuscript "Artificial intelligence on food vulnerability: future implications within a framework of opportunities and challenges", I have a few comments.
Line 5: It would be helpful to briefly explain "the lens of Stephen Hawking" in the introduction, so the reader knows what it refers to.
Line 19: Please remove Fanzo.
Line 178: Please remove Haelein and Kaplan.
Line 203: Also, delete Soori et al.
Lines 205-208: Could you review the writing style in this section?
Line 451: Please specify the source of table 1.
The study explores the challenges and opportunities of AI in different fields, with a focus on global food reality (Lines 8-9). However, the content remains quite general with little emphasis on food issues. The conclusion, while not mentioning food issues, fits well with the rest of the manuscript. In the conclusion (lines 526-528), the authors discuss "the multifaceted nature of AI, its impact on various sectors," which aligns with the content of the manuscript. Therefore, I suggest that the authors remove "Food" from the title and come up with a title that reflects AI's impact on society in general.
Response:
Dear reviewer, thank you very much for your comments, your remarks were addressed, they can be seen in the manuscript highlighted in red. The introduction was improved by expanding the explanation of Stephen Hawking's position. The three references mentioned in your comments were removed. On your last comment about removing the word food in the title, I disagree, however, to support my position I improved the conclusion by including aspects of food security.